# Effect of Peroneus Longus Muscle Release on Abductor Hallucis Muscle Activity and Medial Longitudinal Arch before Toe-Tap Exercise in Participants with Flexible Pes Planus

**DOI:** 10.3390/healthcare10010044

**Published:** 2021-12-27

**Authors:** Youngkyu Choi, Jihyun Lee

**Affiliations:** Department of Physical Therapy, Baekseok University, Cheonan-si 31065, Korea; dudrb5454@naver.com

**Keywords:** fibularis longus, flatfoot, foot pathology, imbalance, soft-tissue therapy

## Abstract

Pes planus is a medical condition of the foot wherein there is a flattening or lowering of the medial longitudinal arch. The abductor hallucis muscle starts at the heel bone and attaches to the medial side of the first toe. Whenever it contracts, it plays a key role in elevating the medial longitudinal arch. Hence, the abductor hallucis muscle should be strong enough to control the depression on the medial longitudinal arch. The peroneus longus muscle plantarflexes the ankle and everts the ankle and subtalar joint. If this muscle contracts more than the abductor hallucis muscle does, the medial longitudinal arch of the foot is depressed. This study aimed to investigate the effect of myofascial release of the peroneus longus before performing the toe-tap exercise for strengthening the abductor hallucis muscle in participants with flexible pes planus. This cross-over study included 16 volunteers with flexible pes planus. The participants performed a toe-tap exercise before and after the myofascial release of the peroneus longus. During the toe-tap exercise, the muscle activity of the abductor hallucis and peroneus longus were measured using a Delsys Trigno Wireless Electromyography System. The angle of the medial longitudinal arch was measured using Image J software. Photos in the sagittal plane were used. The peroneus longus activity and medial longitudinal arch angles were significantly decreased. On the other hand, the activity of the abductor hallucis significantly increased after the myofascial release of the peroneus longus before performing the toe-tap exercise (*p* < 0.05). Individuals with flexible pes planus should be encouraged to perform myofascial release of the peroneus longus before the toe-tap exercise to improve the abductor hallucis activity and to elevate the medial longitudinal arch.

## 1. Introduction

Pes planus is a condition found in approximately 15%–25% of the human population. It is a foot deformity caused by an abnormally low or complete loss of the medial longitudinal arch [1,2,3]. Pes planus is broadly divided into rigid and flexible types. The former is a state of reduced medial longitudinal arch regardless of weight bearing. On the other hand, the latter is characterized by a normal medial longitudinal arch if there is no weight load and an abnormal medial longitudinal arch if there is weight support [4]. The cause of pes planus remains unclear. However, the condition is associated with instability of the first metatarsus or talus, excessive eversion of the subtalar joint, eversion of the rear foot, and abduction of the mid-foot against the rear foot [5,6,7,8]. Pes planus may cause patellofemoral pain syndrome [6], tendinous synovitis of the tibialis posterior [9], or overuse syndrome, including plantar fasciitis [5,10]. The clinical symptoms of pes planus include pain in the plantar fascia and sudden fatigue due to a decrease in shock absorption in the foot during walking or running [11,12]. In addition, during gait or weight support, excessive eversion of the subtalar joint causes a medial shift of weight [13] and deficits in balance. The reduced balance leads to deteriorated stability of the entire body, which reduces stamina upon long hours of walking [1].

To alleviate such symptoms, various treatments are applied in clinical practice, including customized foot orthoses based on the dynamic foot behavior [14], and active strengthening exercises. Foot orthoses are one of the most common interventions in restoring foot kinematics in pes planus. They restore the subtalar joint to a neutral position, and reduce rear foot eversion and forefoot abduction [13,15]. According to two studies, the shape of the orthosis differs per patient, depending on the dynamic foot behavior during gait [14,16]. Individuals with pes planus with excessive rear foot eversion or very flexible medial arches require support on the medial side of the foot, while those with excessive forefoot abduction require support on the lateral side [14]. Active strengthening exercises target the abductor hallucis muscle. The abductor hallucis muscle starts at the heel bone and attaches to the medial side of the first toe. It plays a key role in elevating the medial longitudinal arch upon its contraction [17,18,19,20,21]. Hence, in numerous studies and clinical practice, exercises that strengthen the abductor hallucis muscle are performed to control the depression of the medial longitudinal arch [19,22,23]. The most widely used strengthening exercises in clinical practice are the short-foot, toe-curl, toe-spread, and toe-tap exercises. In a previous study comparing the abductor hallucis activity and the medial longitudinal arch angle during a strengthening exercise, significant improvements in the abductor hallucis activity and the medial longitudinal arch angle were obtained after the toe-tap exercise compared with those after the short-foot or toe-curl exercises [23,24]. The toe-tap exercise induces abduction and flexion of the first metatarsophalangeal joint, while the sole of the foot is resting on an inclined wooden board. This action is the original function of the abductor hallucis muscle.

While various attempts have been made to increase the activity of the abductor hallucis muscle, hyperactivity of the peroneus longus muscle should simultaneously be prevented. This is because the peroneus longus leads to excessive eversion of the subtalar joint, which causes the medial longitudinal arch to collapse. However, no study has investigated the effect of reducing peroneus longus activity while enhancing the abductor hallucis activity in patients with pes planus. The methods applied in clinical practice to reduce hyperactivity of the peroneus longus muscles are stretching, connective tissue massage, and the muscle energy technique. Among them, the fascia and muscle relaxation methods using a foam roller have gained much interest as self-myofascial release methods based on the pressure imparted by their own weights [24].

This study aimed to determine the effects of the peroneus longus muscle release before the toe-tap exercise on the abductor hallucis and peroneus longus muscles and medial longitudinal arch of patients with pes planus using a foam roller, and to compare the effects of peroneus longus muscle release with subsequent toe-tap exercise with the effects of the toe-tap exercise alone. We hypothesized that myofascial release of the peroneus longus muscle before the toe-tap exercise decreased the peroneus longus muscle activity and increased the abductor hallucis muscle activity, which caused the elevation of the medial longitudinal arch.

## 2. Materials and Methods

### 2.1. Participants

G*power software (ver. 3.1.6; Franz Faul, Kiel University, Kiel, Germany) was used to calculate the sample size. By using data obtained from a pilot study of five participants, we calculated the necessary sample size to be 6 participants to achieve a power of 0.80, an effect size of 1.30 (calculated from differences in the mean and standard deviation of the pilot study), and an α level of 0.05. The study included 16 patients with flexible pes planus. Flexible pes planus among the participants was verified using the navicular drop test using a 20-cm ruler. The participants were asked to sit comfortably on a chair with the knees bent at 90°. Subsequently, the knee on the testing side and the second toe were positioned in a straight line. The rater marked the prominent part of the navicular bone using a black marker after manual examination and measured the distance of the mark from the floor. The participants were then asked to stand with both legs providing weight support. The rater measured the distance between the floor and the mark on the navicular bone. The difference between the distance measured in a sitting position and that measured in a standing position was taken as the length of the navicular drop [25]. The navicular drop test was performed twice, and the average value was used. The participants showing means ≥9 mm were identified as having flexible pes planus [26,27]. Individuals who currently had the following or had a history of these were excluded: inflammatory arthritis, operation of the foot or ankle, diabetes, and those who had performed a high-intensity muscular strengthening activity on the previous day. The principal investigator explained the overall flow of the study to each participant, and the study was conducted with the consent of each participant. The investigation was approved by the Institutional Review Board of the Baekseok University in the Republic of Korea (BUIRB-202108-HR-028). Table 1 presents the general characteristics of the participants.

### 2.2. Measurements

#### 2.2.1. Electromyography of Abductor Hallucis and Peroneus Longus Activity

To measure the abductor hallucis and peroneus longus activity, a wireless surface electromyography (EMG) device (Delsys Trigno wireless EMG system, Delsys, Boston, MA, USA) was used. The collected data were analyzed using electromyography software (EMGworks Analysis–Delsys, Boston, MA, USA). The sampling rate was 1 kHz. To eliminate the noise, a 20-Hz low-pass filter and a 500-Hz high-pass filter were used. The signals for the abductor hallucis and peroneus longus muscles were treated as the root of the mean square. To ensure optimum electrical conductivity prior to data collection, the skin resistance was reduced by attaching the skin area to the electrode, which was cleaned with alcohol. The surface EMG sensor on the abductor hallucis muscle was attached 1–2 cm below the rough surface of the navicular bone [23]. The peroneus longus muscle was attached to the top 75% point along the line connecting the fibular head and the lateral talus. To normalize the EMG signals for the abductor hallucis and peroneus longus muscles, manual muscle testing was used to measure the maximal voluntary isometric contraction (MVIC). For the MVIC of the abductor hallucis muscle, the rater clasped the participants’ first toes on the medial and plantar sides, while the participants maintained a neutral heel position. The participant then performed maximum abduction of the first toe and plantar flexion against resistance [23]. To determine the MVIC of the peroneus longus muscle, the participants were asked to perch on a chair with a neutral ankle position. One hand was used to stabilized the ankle area just above the talus. The hand that imparted resistance was placed on the lateral and plantar sides of the mid-foot. The resistance direction was inversion with slight dorsiflexion. Resistance was applied, while the participants performed foot eversion with plantar flexion using maximum force [22]. The MVIC was measured three times, and the mean of the three measurements was used in the data analysis. The abductor hallucis and peroneus longus activity during exercise was measured for 5 s during the toe-tap exercise. Only the middle 3 s were used in the data analysis. The first and last seconds were removed. The mean of the triplicate measurements was used in the analysis. The mean muscle activity for each intervention was quantified as %MVIC [23].

#### 2.2.2. Measurement of the Medial Longitudinal Arch

To measure the medial longitudinal arch angles, photos were taken in the sagittal plane for the analysis using Image J software (National Institutes of Health, Bethesda, MD, USA). The participants were asked to sit on a chair with both feet on the floor, while the rater positioned their ankles and subtalar joints in a neutral position and marked the medial side of the heel bone, the medial side of the first metatarsus head, and the prominent part of the navicular bone with a black marker [23]. The attachment points on the medial side of the heel bone varied depending on the sex of the participant. The attachment point for men was the point that joined the area 40 mm from the rearmost side of the heel bone and the area 35 mm from the floor, while that for women was the point that joined the area 30 mm from the rearmost side of the heel bone and the area 30 mm from the floor. These attachment points were marked using a black marker [23]. Once the marking of the three points was complete, a tripod was placed on a point 50 cm from the medial side of the foot. The smartphone lens was set parallel to the floor, while the smartphone camera (Galaxy S10, made in Korea, Samsung) was placed on the same line as the prominent part of the navicular bone. To minimize electrical interference during photography, the smartphone was set to flight mode. The photos were taken three times in the middle of the exercise, 3 s after the exercise began. Upon completion, Image J software was utilized to analyze the angle of the medial longitudinal arch and the central angle of the line connecting the medial side of the first metatarsal head, the rough side of the navicular bone, and the medial side of the heel bone (Figure 1). The mean of the three measurements of the medial longitudinal arch angles using three photos was used in the data analysis. A decrease in the medial longitudinal arch angle indicates an elevation of the medial longitudinal arch [23].

### 2.3. Exercise Methods

#### 2.3.1. Toe-Tap Exercise

The participants were asked to straighten their backs and sit on a chair with the knee and ankle joints at 90° and their feet on a wooden plate (30 × 20 × 1.5 cm). Styrofoam was placed underneath the wooden plate from the metatarsophalangeal joint to the onset of the heel bone to create a slope for slight dorsiflexion of the ankle. A thin piece of styrofoam (20 × 10 × 0.5 cm), instead of a wooden plate, was placed from the metatarsophalangeal joint to the toe for mid-level flexion of the first metatarsophalangeal joint. To assist the abduction of the first toe, a small pin was placed on the medial side of the bone of the first toe to guide it medially. When everything was set, the participant was guided to perform the metatarsophalangeal joint plantar flexion without any movement of the ankle or the joints of the first toe, while allowing the first toe to push the pin at maximum abduction [23]. The rater monitored whether the subtalar joint maintained a neutral position by checking the joint flexion of the first toe during exercise. The values obtained when the first toe joint was flexed or when the subtalar joint escaped the neutral position were removed from the final analysis [23].

#### 2.3.2. Myofascial Release of the Peroneus Longus Muscle Using a Foam Roller

A cylindrical foam roller made using a synthetic material composed of ethylene and vinyl acetate (10 cm diameter, 30 cm length; Plus ‘O’ Minus Fitness Products (Suzhou) Co., Ltd., Suzhou, China) was used for the three sets of the exercise, with 2 min myofascial release exercise and 1 min resting time as a single set. The participants were asked to squat (Figure 2A) and abduct the hip joint of the side being tested. The hip joint was then laterally rotated, thereby resulting in a cross-legged position, so that the peroneus longus muscle could be placed on the foam roller (Figure 2B). The participants were guided to clasp the area above the ankle using the hand on the testing side (Figure 2C). To prevent unnecessary movements of the ankle or tibia, the participants were guided to clasp the lateral talus using the second and fourth fingers on the free side and to clasp the medial side of the heel bone using the first finger (Figure 2D). The myofascial release was performed as the participants let the lower arm on the testing side touch the medial fibula and performed loaded movements along the direction of the peroneus longus muscle (Figure 2E). 

### 2.4. Data analysis

All statistical analyses were performed using SPSS for Windows (ver. 18.0, SPSS, Inc., Chicago, IL, USA). A normality test was performed using the Kolmogorov–Smirnov test. The data were normally distributed. The parameter test was then performed. To compare the changes in the abductor hallucis and peroneus longus activity and the medial longitudinal arch angles between the toe-tap exercise alone and the myofascial release of the peroneus longus muscle prior to the toe-tap exercise, the paired t-test was used. Data were expressed as mean ± S.D. The level of significance was set to 0.05. The effect sizes were calculated using Cohen’s d to determine meaningful changes. Cohen’s d is defined as differences between the mean of the toe-tap exercise only and myofascial release of the peroneus longus before the toe-tap exercise divided by SD of the toe-tap exercise only, where an effect size of ≤0.10 indicates a very small change; 0.20, a small change; 0.50, a moderate change; 0.80, a large change; 1.20, a very large change; and 2.0, a huge change [28]. 

## 3. Results

### 3.1. The Abductor Hallucis and Peroneus Longus Activity

Compared to the toe-tap exercise alone, myofascial release of the peroneus longus muscle prior to the toe-tap exercise led to a significant increase in the abductor hallucis activity and a significant decrease in the peroneus longus activity (*p* < 0.05) (Table 2).

### 3.2. The Medial Longitudinal Arch Angles

Compared to the toe-tap exercise alone, myofascial release of the peroneus longus muscle before the toe-tap exercise led to a significant decrease in the medial longitudinal arch angles (*p* < 0.05). A decrease in the medial longitudinal arch angle indicates elevation of the medial longitudinal arch (Table 2).

## 4. Discussion

Pes planus is a medical condition of the foot characterized by flattening or lowering of the medial longitudinal arch. The abductor hallucis muscle starts at the heel bone and attaches to the medial side of the first toe. It plays a key role in the elevation of the medial longitudinal arch upon its contraction. Hence, exercises to strengthen the abductor hallucis muscle are utilized to control the depression of the medial longitudinal arch. On the other hand, the peroneus longus muscle plantarflexes the ankle and everts the ankle and subtalar joint. An even greater contraction of this muscle as compared to that of the abductor hallucis results in the depression of the medial longitudinal arch of the foot. This study aimed to determine the effects of the release of the peroneus longus muscle prior to the toe-tap exercise on the abductor hallucis and peroneus longus activity and medial longitudinal arch in patients with flexible pes planus. The results showed that performing the toe-tap exercise after myofascial release of the peroneus longus muscle significantly increased the abductor hallucis activity, decreased the peroneus longus activity, and elevated the medial longitudinal arch.

The abductor hallucis muscle activity was increased by 27.57%, while the medial longitudinal arch was improved by 2.34% after the myofascial release of the peroneus longus muscle before the toe-tap exercise. These results can be explained by the succeeding sentences. First, an excessive contraction of the peroneus longus muscle results in the eversion of the subtalar joint to suppress the abductor hallucis muscle that supports the medial longitudinal arch. Hence, the relaxation of the peroneus longus muscle leads to the elevation of the medial longitudinal arch and the reduction of the potential muscle imbalance to increase the activity of the abductor hallucis muscle, while the medial longitudinal arch is elevated before the toe-tap exercise. According to previous studies involving different participants and exercise methods, the lower trapezius activity was significantly increased, while the round shoulder posture was significantly decreased after pectoralis minor stretching before scapula posterior depression [29]. Another study reported that the tibialis anterior activity and the ankle joint dorsiflexion angle were significantly increased after gastrocnemius stretching before resistance exercise in participants with limited dorsiflexion of the ankle joint [30]. In addition, a study showed that the rectus femoris activity was significantly increased after the use of a foam roller through the relaxation of the hamstrings [31].

Upon the excessive contraction of the peroneus longus muscle, the medial longitudinal arch collapses due to the increased eversion of the subtalar joint. In contrast, the relaxation of the peroneus longus muscle increases the range of the elevation of the medial longitudinal arch by increasing the inversion of the subtalar joint. The extended range enables the abductor hallucis muscle to contract more easily and continuously. Third, the peroneus longus contraction does not only evert the subtalar joint but also plantarflexes the foot. It was reported in a previous study that the activity of the plantar flexor muscle increases against the ground reaction force during the abductor hallucis muscle exercise, as daily activities do not often require selective movements of the first toe [32,33]. The plantar flexor activity against the ground reaction force during the toe-tap exercise is likely to decrease as the activity of the peroneus longus muscle decreases after the myofascial release of the peroneus longus muscle. The decrease in the compensation reaction may, thus, increase the pure muscle activity of the abductor hallucis. In light of the findings in this study, myofascial release of the peroneus longus muscle before the toe-tap exercise is recommended for patients with flexible pes planus. Myofascial release increases the abductor hallucis activity and enhances the medial longitudinal arch, thereby elevating it.

The myofascial release of the peroneus longus muscle before the toe-tap exercise reduced the peroneus longus activity by 50.90%. Since a foam roller was used in the relaxation of the peroneus longus muscle, the Golgi tendon organ, a proprioceptor, was stimulated. The resulting autogenic inhibition lowered the motor unit excitation to decrease the muscle tension [34]. In addition, during the process of muscle relaxation, the muscle tone was probably decreased as the mechanical receptors Ruffini and Pacinian corpuscles exerted an inhibitory effect on the central nervous system [35]. Thus, the myofascial release of the peroneus longus muscle using a foam roller in this study reduced the peroneus longus activity, which likely prevented excessive foot eversion in patients with flexible pes planus.

There were several limitations in this study that should be addressed in future studies. First, although the participants were given enough resting time, repeated use of the abductor hallucis muscle could have caused muscle fatigue. Therefore, in the future, adequate exercise time and number of sessions that do not induce muscle fatigue in the abductor hallucis muscle should be investigated. Second, during the toe-tap exercise, it was difficult to prevent the isometric contraction of the plantar flexor muscle in the ankle joint, despite efforts to maintain a neutral subtalar joint position without ankle plantar flexion. Upon performing the maximum toe flexion exercise with a constant neutral ankle joint position in a previous study, the muscle activity of the extrinsic plantar flexor (gastrocnemius and soleus muscles) was increased [32]. However, the muscle activity of the plantar flexor was not measured in this study. Thus, in subsequent studies, the muscle activity of the plantar flexors, in addition to that of the abductor hallucis and peroneus longus, should be measured. Third, only the short-term effect of the peroneus longus muscle release before the toe-tap exercise was examined in this study. Thus, the long-term effect of peroneus longus muscle release before the toe-tap exercise should also be determined in the future. Finally, this study selected asymptomatic individuals with pes planus. If there was pain or discomfort, the results of the study may have been different. For further clinical applications, future studies should include symptomatic patients with pes planus to find out how they respond to function and pain. 

## 5. Conclusions

This study investigated the effects of peroneus longus muscle release before the toe-tap exercise for strengthening abductor hallucis activity, decreasing peroneus longus activity, and elevating the medial longitudinal arch. The study findings suggest that myofascial release of the peroneus longus muscle should be considered in designing a training program to enhance abductor hallucis activity and elevate the medial longitudinal arch in patients with flexible pes planus.

## Figures and Tables

**Figure 1 healthcare-10-00044-f001:**
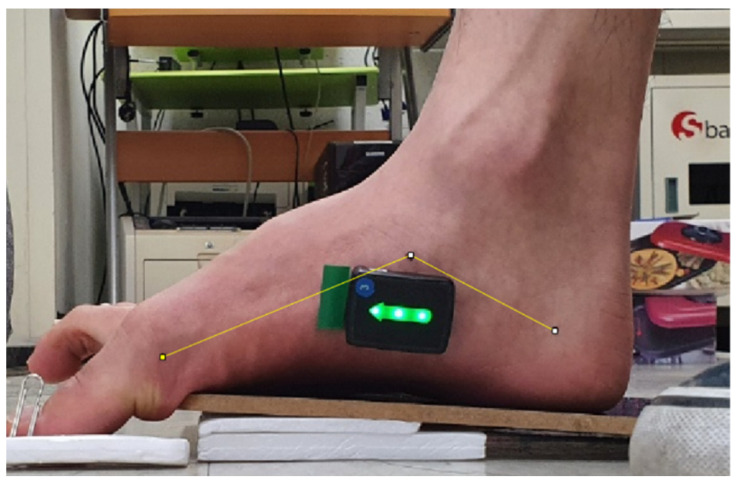
Measurement of the medial longitudinal arch.

**Figure 2 healthcare-10-00044-f002:**
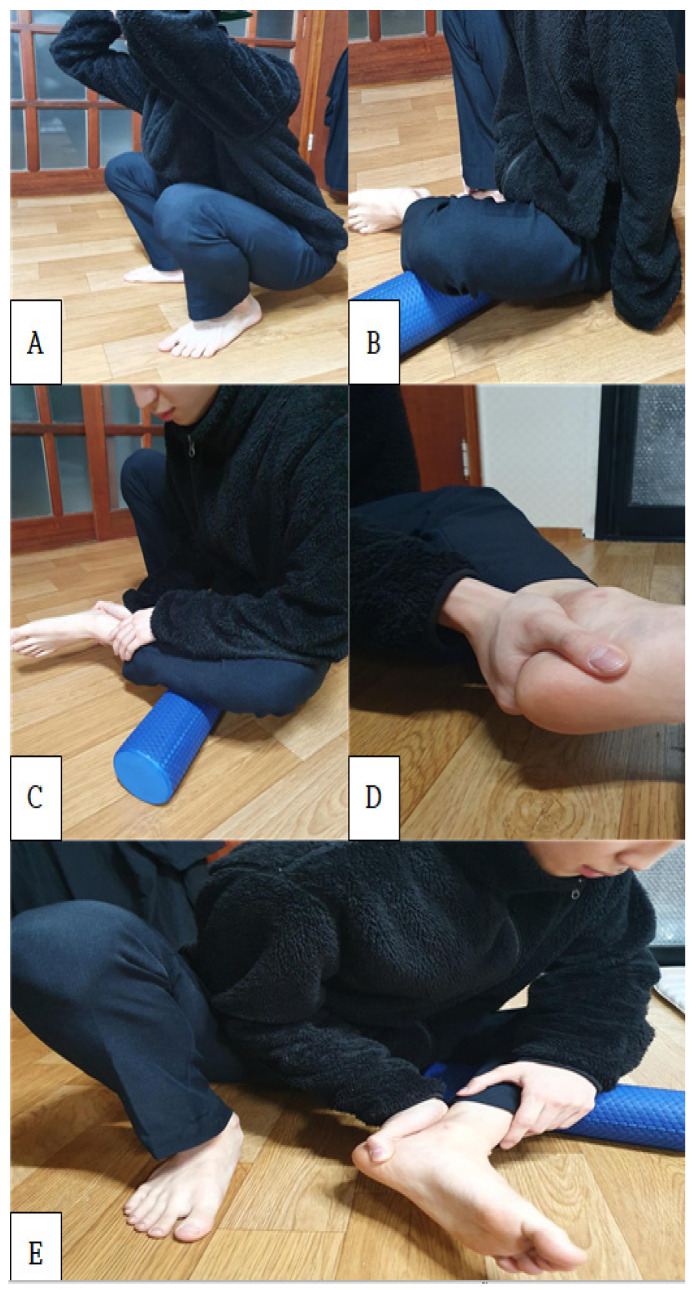
Myofascial release of the peroneus longus muscle. (**A**) squat, (**B**) hip abduction and lateral rotation, the peroneus longus muscle was placed on the foam roller, (**C**) clasp the area above the ankle, (**D**) clasp the lateral talus (**E**) myofascial release of the peroneus longus muscle.

**Table 1 healthcare-10-00044-t001:** General characters.

Variables	Mean	±	SD ^a^
Age (yrs)	23.18	±	2.07
Height (cm)	170.18	±	9.20
Weight (kg)	67.06	±	20.32
Navicular drop test (cm)	1.20	±	0.28
Gender	10 males		6 females

^a^ Mean ± standard deviation.

**Table 2 healthcare-10-00044-t002:** Comparison of the muscle activity and medial longitudinal arch angles between the toe-tap exercise alone and the myofascial release of the peroneus longus with subsequent toe-tap exercise.

Variables	Toe-Tap Exercise Only ^a^	Myofascial Release of the Peroneus Longus before Toe-Tap Exercise ^a^	Differences ^a^	Effect Size	*p*
Abductor hallucis (%MVIC)	50.75 ± 21.42	64.74 ± 25.95	13.98 ± 10.77	0.65	<0.001 *
Peroneus longus (%MVIC)	19.27 ± 10.76	12.77 ± 9.24	6.5 ± 5.35	0.60	<0.001 *
Medial longitudinal arch (°)	133.30 ± 5.56	130.25 ± 6.02	3.04 ± 2.03	0.55	<0.001 *

^a^ Mean ± standard deviation. %MVIC: percent of maximal voluntary isometric contraction. * *p* > 0.05. A decrease in the medial longitudinal arch angle indicates an elevation of the medial longitudinal arch.

## Data Availability

The data presented in this study are available upon request from the corresponding author.

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
