# Peer review of "Effect of Peroneus Longus Muscle Release on Abductor Hallucis Muscle Activity and Medial Longitudinal Arch before Toe-Tap Exercise in Participants with Flexible Pes Planus"

_healthcare, 2021, doi:10.3390/healthcare10010044_

Round 1
Reviewer 1 Report
The methodology of the study is well described and appropriate.
I recommend using “myofascial release” or similar in place of “release” in the abstract and throughout the manuscript. As I read the abstract, the wording sounds like a surgical release.
Abstract: Background should mention flatfoot somehow.
The sentence “The participants performed a toe tap exercise only and released the peroneus longus before the toe tap exercise” would read clearer as “The participants performed a toe tap exercise before and after myofascial release of the peroneus longus”.
At the beginning of the discussion, I recommend a statement of why the abductor hallucis was selected given it’s important to the medial longitudinal arch to reorient the reader why that muscle was analyzed.
Recommend stating the medial longitudinal arch height increased rather than was “enhanced” in the first paragraph.
Results section 3.2: Either here or in the table, it should be conveyed that a decrease in arch angle = an increase in the medial longitudinal arch because it’s a little counterintuitive at first glance.
Were any of the participants symptomatic and was pain assessed? If it wasn’t included in this study, this warrants a discussion point for future work to determine if the improvements in arch height and muscle activity translate to functional or pain improvements.
Long paragraph on page 6: This paragraph is too long and need to be broken up into more focused points.
Table 2: List as p<0.001 not p=0
Author Response
Thank you for your thoughtful review.
I recommend using “myofascial release” or similar in place of “release” in the abstract and throughout the manuscript. As I read the abstract, the wording sounds like a surgical release.
- We changed “release” to “myofascial release”.
Abstract: Background should mention flatfoot somehow.
- We added some sentences about the research background.
The sentence “The participants performed a toe tap exercise only and released the peroneus longus before the toe tap exercise” would read clearer as “The participants performed a toe tap exercise before and after myofascial release of the peroneus longus”.
- We replaced it with the sentence you corrected. We appreciated it.
Results section 3.2: Either here or in the table, it should be conveyed that a decrease in arch angle = an increase in the medial longitudinal arch because it’s a little counterintuitive at first glance.
- We agreed with your opinion. We added it in the Results section 3.2 and the table .
Table 2: List as p<0.001 not p=0
- We changed “p=0” to “p<0.001”.
At the beginning of the discussion, I recommend a statement of why the abductor hallucis was selected given it’s important to the medial longitudinal arch to reorient the reader why that muscle was analyzed.
- We added some sentences about the abductor hallucis at the beginning of the discussion.
Recommend stating the medial longitudinal arch height increased rather than was “enhanced” in the first paragraph.
- We changed “enhanced” to “increased”.
Were any of the participants symptomatic and was pain assessed? If it wasn’t included in this study, this warrants a discussion point for future work to determine if the improvements in arch height and muscle activity translate to functional or pain improvements.
- We totally agreed with your opinion. We added it in the discussion section.
Long paragraph on page 6: This paragraph is too long and need to be broken up into more focused points.
- We have divided the paragraph.

Reviewer 2 Report
This manuscript entitled “Effect of Peroneus Longus Muscle Release before Toe Tap Exercise on the Abductor Hallucis Muscle Activity and Medial Longitudinal Arch in Participants with Flexible Pes Planus” primarily aimed to investigate the effect of release of the peroneus longus before performing toe tap exercise for strengthening the abductor hallucis muscle in participants with flexible pes planus. This work is well written and the topic will be of great interest to the readership. There are however some aspects to be improved before this manuscript can be accepted for publication, which lists below.
Specific comments
- Abstract, it is not enough to introduce peroneus longus muscle in the background. It is recommended to further strengthen the connection with the research purpose in this section.
- Please modify and improve the quality of the keywords as this will assist others when they are searching for information on your research topic.
- Introduction, it is suggested that the authors should present more fields that would involve foot deformities represented by flat feet, which could further help to highlight the practical meaning of this study. For example, here, ‘To ameliorate such symptoms, various treatments are applied in clinical practice’, several latest articles can be provided to the authors:
- A current review of foot disorder and plantar pressure alternation in the elderly[J]. Physical Activity and Health, 2020, 4(1).
- Understanding the role of foot biomechanics on regional foot orthosis deformation in flatfoot individuals during walking[J]. Gait & Posture, 2022, 91: 117-125.
- Arch-Support Induced Changes in Foot-Ankle Coordination in Young Males with Flatfoot during Unplanned Gait Termination[J]. Journal of Clinical Medicine, 2021, 10(23): 5539.
- ‘… significant improvements in the abductor hallucis muscle activity and the medial longitudinal arch angle were obtained from the toe tap exercise…’, more details about the potential effects and mechanisms of toe tap exercise need to be provided, which will help highlight the necessity of this research.
- ‘The study included 16 patients with flexible pes planus’, the sample size would affect the result of statistical analysis. The authors should illustrate that whether the sample size of this trial was calculated or just be set according to experience?
- How many navicular drop tests were performed by each subject? Test errors may need to be considered.
- Please report the gender of the subject in this study in Table 1.
- Results, please provide objective data instead of simple text and P-value. I suggest that the authors can present the results of this study in the form of tables or figures (better), and provide the main data in the text.
- Discussion, it is suggested that this part should focus more on the comparison with previous related studies.
- ‘As a foam roller was used in the relaxation of the peroneus longus muscle, the Golgi tendon organ as the proprioceptor is likely to have been …’, please add a ref here.
- Conclusion, please avoid re-emphasizing results. This part should also be further strengthened based on the main findings of this study.
- In summary, please make sure that your manuscript is properly prepared (without any grammatical and spelling mistakes) and formatted before submitting a revision.
Author Response
Thank you for your thorough review.
Abstract, it is not enough to introduce peroneus longus muscle in the background. It is recommended to further strengthen the connection with the research purpose in this section.
- We added some sentences about the research background.
Please modify and improve the quality of the keywords as this will assist others when they are searching for information on your research topic.
- We changed the keywords that did not overlap with the title as a Mesh terms.
Introduction, it is suggested that the authors should present more fields that would involve foot deformities represented by flat feet, which could further help to highlight the practical meaning of this study. For example, here, ‘To ameliorate such symptoms, various treatments are applied in clinical practice’, several latest articles can be provided to the authors:
A current review of foot disorder and plantar pressure alternation in the elderly[J]. Physical Activity and Health, 2020, 4(1).
Understanding the role of foot biomechanics on regional foot orthosis deformation in flatfoot individuals during walking[J]. Gait & Posture, 2022, 91: 117-125.
Arch-Support Induced Changes in Foot-Ankle Coordination in Young Males with Flatfoot during Unplanned Gait Termination[J]. Journal of Clinical Medicine, 2021, 10(23): 5539.
- Thanks for recommending good articles. We have carefully reviewed the articles and added it to the introduction section.
‘… significant improvements in the abductor hallucis muscle activity and the medial longitudinal arch angle were obtained from the toe tap exercise…’, more details about the potential effects and mechanisms of toe tap exercise need to be provided, which will help highlight the necessity of this research.
- We added more details about mechanisms of toe tap exercise.
‘The study included 16 patients with flexible pes planus’, the sample size would affect the result of statistical analysis. The authors should illustrate that whether the sample size of this trial was calculated or just be set according to experience?
- We added the method of sample size calculation.
How many navicular drop tests were performed by each subject? Test errors may need to be considered.
- The navicular drop test was performed twice, the average value was used. We added the sentence.
Please report the gender of the subject in this study in Table 1.
- We reported the gender of the subject in this study in Table 1.
Results, please provide objective data instead of simple text and P-value. I suggest that the authors can present the results of this study in the form of tables or figures (better), and provide the main data in the text.
- We added Table 2.
Discussion, it is suggested that this part should focus more on the comparison with previous related studies.
- We have agreed with your opinion. Since it is a new concept in pes planus, it was difficult to find similar researches. Thus, we compared studies with similar concepts but not the same subjects.
‘As a foam roller was used in the relaxation of the peroneus longus muscle, the Golgi tendon organ as the proprioceptor is likely to have been …’, please add a ref here.
- We added the reference.
Conclusion, please avoid re-emphasizing results. This part should also be further strengthened based on the main findings of this study.
- We revised the conclusion.
In summary, please make sure that your manuscript is properly prepared (without any grammatical and spelling mistakes) and formatted before submitting a revision.
- We have received English proofreading and attached a certificate.

Round 2
Reviewer 2 Report
Most of my questions have been well addressed, I am satisfied with current version, recommending accept now.
Author Response
I really appreciate it